# Evaluation of the Diagnostic Performance of Two Automated SARS-CoV-2 Neutralization Immunoassays following Two Doses of mRNA, Adenoviral Vector, and Inactivated Whole-Virus Vaccinations in COVID-19 Naïve Subjects

**DOI:** 10.3390/microorganisms11051187

**Published:** 2023-04-30

**Authors:** Eszter Csoma, Ágnes Nagy Koroknai, Renáta Sütő, Erika Szakács Szilágyi, Marianna Pócsi, Attila Nagy, Klára Bíró, János Kappelmayer, Béla Nagy

**Affiliations:** 1Department of Medical Microbiology, Faculty of Medicine, University of Debrecen, Nagyerdei krt. 98, 4032 Debrecen, Hungary; csoma.eszter@med.unideb.hu; 2Department of Laboratory Medicine, Faculty of Medicine, University of Debrecen, Nagyerdei krt. 98, 4032 Debrecen, Hungary; koroknai.agness@gmail.com (Á.N.K.); sutorenata@gmail.com (R.S.); szakacsne.erika@gmail.com (E.S.S.); pmarcsi89@gmail.com (M.P.); kappelmayer@med.unideb.hu (J.K.); 3Intensive Care Unit, Gyula Kenézy Campus, University of Debrecen, Bartók Béla út 2-26, 4031 Debrecen, Hungary; 4Doctoral School of Kálmán Laki, Faculty of Medicine, University of Debrecen, Nagyerdei krt. 98, 4032 Debrecen, Hungary; 5Department of Health Informatics, Institute of Health Sciences, Faculty of Health, University of Debrecen, Kassai út 26, 4028 Debrecen, Hungary; nagy.attila@sph.unideb.hu; 6Institute of Health Economics and Management, Faculty of Economics and Business, University of Debrecen, Nagyerdei krt. 98, 4032 Debrecen, Hungary; kbiro@med.unideb.hu

**Keywords:** SARS-CoV-2, serology, vaccine, different vaccine types, COVID-19 disease, neutralizing antibody

## Abstract

Background: Limited data are available on humoral responses determined by automated neutralization tests following the administration of the three different types of COVID-19 vaccinations. Thus, we here evaluated anti-SARS-CoV-2 neutralizing antibody titers via two different neutralization assays in comparison to total spike antibody levels. Methods: Healthy participants (*n* = 150) were enrolled into three subgroups who were tested 41 (22–65) days after their second dose of mRNA (BNT162b2/mRNA-1273), adenoviral vector (ChAdOx1/Gam-COVID-Vac) and inactivated whole-virus (BBIBP-CorV) vaccines, with no history or serologic evidence of prior SARS-CoV-2 infection. Neutralizing antibody (N-Ab) titers were analyzed on a Snibe Maglumi^®^ 800 instrument and a Medcaptain Immu F6^®^ Analyzer in parallel to anti-SARS-CoV-2 S total antibody (S-Ab) levels (Roche Elecsys^®^ e602). Results: Subjects who were administered mRNA vaccines demonstrated significantly higher SARS-CoV-2 N-Ab and S-Ab levels compared to those who received adenoviral vector and inactivated whole-virus vaccinations (*p* < 0.0001). N-Ab titers determined by the two methods correlated with each other (r = 0.9608; *p* < 0.0001) and S-Ab levels (r = 0.9432 and r = 0.9324; *p* < 0.0001, respectively). Based on N-Ab values, a new optimal threshold of Roche S-Ab was calculated (166 BAU/mL) for discrimination of seropositivity showing an AUC value of 0.975 (*p* < 0.0001). Low post-vaccination N-Ab levels (median value of 0.25 μg/mL or 7.28 AU/mL) were measured in those participants (*n* = 8) who were infected by SARS-CoV-2 within 6 months after immunizations. Conclusion: Both SARS-CoV-2 N-Ab automated assays are effective to evaluate humoral responses after various COVID-19 vaccines

## 1. Introduction

As of the end of March 2023, more than 761 million people have been infected with SARS-CoV-2 virus worldwide and over 6.8 million patients have died of COVID-19 disease (https://covid19.who.int, accessed on 25 March 2023). Although vaccines have remained crucial for preventing hospitalization and death serving a valuable immune-priming potential for unvaccinated individuals, there are limited data and there is still a debate on what type(s) of vaccine and which (booster) vaccination strategy, i.e., homologous, or heterologous immunization can provide more effective humoral responses [1,2,3,4,5]. When COVID-19 vaccines became available in Hungary in the beginning of 2021, the government established a risk-based vaccination program for the COVID-19 vaccines, because the first available BNT162b2 mRNA vaccine was not available for all inhabitants. The principle of four priority target groups was founded regarding the urgent need for vaccination: (1) healthcare and social care workers; (2) individuals with COVID-19 associated risks (e.g., over 60 years of age, or age between 18–59 years with underlying comorbidities); (3) those working in critical infrastructures, e.g., at schools; and (4) people between the ages of 18–59 years who are not included in any of these groups (Plan of tasks related to COVID-19 vaccination; in Hungarian: http://koronavirus.gov.hu, accessed on 25 January 2021). Accordingly, the BNT162b2 mRNA vaccine was administered to all healthcare professionals and social workers, while individuals in other risk groups were immunized at a large ratio with various other types of COVID-19 vaccines, such as adenoviral vector-based (ChAdOx1 or Gam-COVID-Vac) and inactivated whole-virus BBIBP-CorV vaccine.

These vaccinations act via distinct mechanisms to induce immune activation: (1) lipid nanoparticle delivery system is used in mRNA vaccines to deliver SARS-CoV-2 S1-specific mRNA [6], while adenoviral vector-based Gam-COVID-Vac and ChAdOx1 vaccines contain non-replicating virus vector as a delivery system for coding S1 protein [7], and BBIBP-CorV consists of whole virion (nucleocapsid, spike and membrane proteins) in inactivated form [8].

In contrast to our local pandemic conditions, the mRNA vaccines were predominantly applied for immunization of healthcare workers [9] and patients with various chronic diseases [10,11] in many countries. Based on a large trial, the mRNA-1273 vaccine showed over 94% efficacy at preventing COVID-19 illness [12]. There have been only a limited number of publications on immunogenicity studies after two doses of COVID-19 vaccine from those countries where different vaccine types (mRNA, adenoviral vector, inactivated) have been used [13,14,15,16,17,18,19,20]. Thus, for the first time, we herein compared the effects of the mRNA, adenoviral vector-based and inactivated whole-virus COVID-19 vaccines via humoral responses in a Hungarian cohort.

In the last two years, new automated COVID-19 neutralization immunoassays have become commercially available, e.g., on the Snibe Maglumi^®^ instruments [21,22] and the Medcaptain Immu F6^®^ Analyzer [23], to measure the neutralizing capacity of anti-SARS-CoV-2 immunoglobulins (Ig). Hence, for the first time, we utilized these two tests for neutralizing antibody (N-Ab) in parallel (i) to measure neutralizing antibody levels after two doses of a wide range of vaccines, and (ii) to compare the results determined by these methods.

## 2. Materials and Methods

### 2.1. Study Participants

We assessed SARS-CoV-2-specific post-vaccination antibody responses in a total of 150 consecutive adult volunteers who received two doses of any of the three different types of homologous vaccinations. These subjects had no known history or serologic evidence of prior SARS-CoV-2 infection based on subsequent RT-PCR tests and measurement of Cobas^®^ anti-SARS-CoV-2 total Ig levels against N-protein (Roche Diagnostics, Mannheim, Germany). Among the volunteers, 50 subjects (35 females and 15 males) at a median age of 61.1 years received mRNA vaccines (*n* = 39 with BNT162b2, Pfizer/BioNTech and *n* = 11 with mRNA-1273, Moderna), 50 individuals (26 females and 24 males, aged 53.6 years) were injected with adenoviral vector vaccines (*n* = 28, ChAdOx1, AstraZeneca and *n* = 22, Gam-COVID-Vac, Sputnik V) and 50 persons (30 females and 20 males with a median age of 63.6 years) were administered with inactivated whole-virus vaccine BBIBP-CorV (Sinopharm) (Table 1). Interestingly, more female participants were finally enrolled into study sub-cohorts, because especially in the early phase of vaccination program, more (COVID-19 naïve) women had applied for any vaccine, and more male patients were excluded from the study due to their previous SARS-CoV-2 infection. Specimens were obtained between 15 February 2021, and 30 May 2021, (median, IQR) 41 (22—65) days after their second dose in the whole population. The exclusion criteria included age < 18 years, known primary immunodeficiency, malignancy, and ongoing immunosuppressive therapy. The study was approved by the Scientific and Research Ethics Committee of the University of Debrecen and the Ministry of Human Capacities (32568–7/2020/EÜIG). Written informed consent was obtained from all participants. The study was performed according to the Declaration of Helsinki.

### 2.2. Laboratory Methods

Venous blood was collected in Vacutainer^®^ tubes, and serum samples were stored at −80 °C before analysis. Anti-SARS-CoV-2 N-Abs were determined on the Snibe quantitative assay (Snibe Maglumi^®^ 800) based on indirect chemiluminescence immunoassay (CLIA) according to the manufacturer’s instructions (Snibe, Shenzhen, China). The measuring range included values between 0.05 and 30 μg/mL, while an automated 2-fold sample dilution extended the range of quantification up to 60 μg/mL if necessary. In parallel, another automated competitive CLIA method was applied to analyze N-Ab levels using an Immu F6^®^ Automatic Chemiluminescent Immunoassay Analyzer (Shenzhen Medcaptain Medical Technology Co., Shenzhen, China) according to the manufacturer’s instructions. The neutralizing antibody level was calculated in AU/mL (arbitrary units per milliliter). The lower and the upper limit of quantification was 3 AU/mL, and 100 AU/mL, respectively. Serum samples above the upper limit were diluted according to the manufacturer’s instructions. As announced by Medcaptain Medical Technology Co., SARS-CoV-2 N-Abs corresponds to WHO standard materials (NIBSC 21/234; https://nibsc.org/documents/ifu/21-234.pdf; accessed on 1 October 2021): conversion unit 1 AU/mL = 29.76 IU/mL (international units per milliliter). For comparison, total SARS-CoV-2 S-specific antibody (S-Ab) titers were quantified by a Cobas^®^ Anti-SARS-CoV-2 S serology test (Roche). The test consisted of an electro-chemiluminescence indirect assay (ECLIA) and included two recombinant RBD antigens, which bound serum antibodies in a double-antigen sandwich setup. The measuring range included values between 0.40 and 250 BAU/mL (binding antibody units per milliliter), but automated sample dilutions (1:200) extended the upper range of quantification up to 50.000 BAU/mL. Total anti-SARS-CoV-2 S-specific antibody results measured in U/mL were converted to the WHO international unit based on a user circular provided by the manufacturer (U/mL = 0.972 × BAU/mL). Seropositivity was evaluated based on the manufacturer’s cut-off values of 0.3 μg/mL (Snibe), 5.0 AU/mL (Medcaptain), and 0.8 BAU/mL (Roche), respectively.

### 2.3. Statistical Analysis

The Shapiro–Wilk and Kolmogorov–Smirnov tests were used for evaluation of the normality of data. In case of non-parametric clinical and laboratory values, the comparison between two groups of independent data was done with Mann–Whitney U-test or Fisher’s exact test. Spearman correlation coefficient (r) was used to explore relationships between N-Ab and total S-Ab levels. The area under the receiver operating characteristic curve (ROC-AUC) value was determined for total S-Ab to indicate seropositivity when the cut-off value of 166 BAU/mL was set as a binary classifier. The maximum of Youden index (sensitivity, 100-specificity) was determined to identify the cut-off values. The ratio of seropositive individuals after two doses was separately calculated in each vaccination cohort based on the results of the two different neutralization tests and S-Ab concentrations using the new cut-off value of Roche test. Inter-rater reliability of serostatus classification was calculated using Cohen’s kappa with 95% confidence intervals with the determination of percentage agreement for different serological tests within each vaccination subgroup and the entire population. The *p* < 0.05 probability value was accepted as significant. Analyses were performed using GraphPad Prism, version 9 (GraphPad Software, La Jolla, CA, USA).

## 3. Results

### 3.1. Comparison of the Levels of Anti-SARS-CoV-2 Neutralizing Antibody by Two Different Neutralization Tests with Total Spike Immunoglobulin following Two Doses of Three Different Vaccinations

First, we evaluated the titer of anti-SARS-CoV-2 N-Abs following two doses of mRNA, adenoviral vector, and inactivated whole-virus vaccines to compare the clinical performance of two different automated neutralization assays via the effectiveness of various immunizations. Using the Snibe assay, the highest concentrations of N-Abs were measured after mRNA vaccines with the median value of 1.59 μg/mL, while significantly lower levels (*p* < 0.0001) were measured in those who received adenoviral-vector-based or inactivated whole-virus vaccinations (0.50 and 0.41 μg/mL, respectively) (Figure 1A). A similar tendency was found among the three vaccination subgroups (31.82, 12.25 and 8.97 AU/mL, respectively) with the Medcaptain test (Figure 1B). Furthermore, these sera were analyzed for total S-Ab levels, and significant alterations (*p* < 0.0001) were observed in these titers (993, 237 and 139 BAU/mL, respectively) among sub-cohorts (Figure 1C). Within the mRNA and adenoviral vector vaccine sub-groups, we separately analyzed whether there was any difference in antibody levels after the two subtypes of mRNA and adenovirus-vector-based immunizations. Although there were relatively higher N-Ab titers (median value 1.13 vs. 3.65 μg/mL; *p* = 0.0603), and (25.1 vs. 52.3 AU/mL; *p* = 0.093) by both neutralization tests as well as total S-Ab levels (692 vs. 1697 BAU/mL; *p* = 0.2104) by the Roche assay in those with mRNA-1273 vs. BNT162b2 mRNA vaccines, these differences were not statistically significant. Similarly, ChAdOx1 vaccinations did not induce significantly larger concentrations of N-Abs (0.56 vs. 0.43 μg/mL; *p* = 0.0906 and 13.85 vs. 11.43 AU/mL; *p* = 0.0940) and S-Abs (248 vs. 179 BAU/mL; *p* = 0.0726) compared with Gam-COVID-Vac (data not shown). Next, we determined if the values of anti-SARS-CoV-2 antibodies correlated with each other in the entire study population. Spearman test was performed that showed a significant linear correlation between Snibe N-Abs and S-Ab levels: r = 0.9432; *p* < 0.0001 (Figure 1D), Medcaptain N-Abs and S-Abs: r = 0.9324; *p* < 0.0001 (Figure 1E) and between the two N-Abs titers: r = 0.9608; *p* < 0.0001 (Figure 1F). These results suggest that mRNA vaccines were the most effective based on induced N-Abs and S-Ab concentrations than adenoviral vector and inactivated whole-virus vaccinations, while the two latter immunizations resulted in a similar degree of antibody responses. In addition, both neutralization assays showed an equal performance to evaluate N-Ab titers.

### 3.2. The Effect of Age, Gender and Elapsed Time between Vaccination and Serology on Neutralizing Antibodies and Total Spike Immunoglobulins

There was no significant difference in age, gender, and the elapsed time between blood sampling for serology and the administration of second dose of vaccines among these sub-cohorts. In addition, the ratio of underlying comorbidities was similar (Table 1). Despite these facts, we investigated whether age and sex could potentially influence the levels of N-Abs and S-Abs using the pooled data of all cohorts. There was no association between age and either N-Ab or S-Ab levels (r = −0.1235; *p* = 0.1321, r = −0.1110; *p* = 0.1762, and r = −0.088; *p* = 0.2836, respectively) (Appendix A). When these titers were compared between female and male subjects, no significant difference was shown (*p* = 0.2334, *p* = 0.0941 and *p* = 0.3534, respectively) (Appendix A). Of note, we pursued to enroll subjects with no difference in the time interval between the second dose of vaccines and serology testing among the sub-cohorts, because as we expected, modest but significant negative associations were found between the elapsed time from the second dose of vaccines and serology testing and the levels of anti-SARS-CoV-2 antibodies (r = −0.2161; *p* = 0.0079 (Snibe), r = −0.2675; *p* = 0.0009 (Medcaptain), r = −0.2293; *p* = 0.0048 (Roche), respectively). Based on these data above, age and gender did not modify the values of N-Abs and S-Ab levels in this study cohort, while the modulatory effect of time interval between serology and vaccination was excluded. Thus, any difference in humoral responses that we observed among the three vaccination sub-groups might be due to the distinct immunogenicity of various vaccines. 

### 3.3. Ratio of Seropositivity after Two Doses of mRNA, Adenoviral Vector, and Inactivated Whole-Virus Vaccinations Based on Two Different Neutralization Tests

After the assessment of single values of N-Abs and S-Abs in all study participants administered with different COVID-19 vaccines, we determined the ratio of seropositive subjects in each cohort to further characterize the effectiveness of vaccine types. Seropositivity was individually evaluated via the results of the two neutralization assays using the manufacturer’s N-Ab cut-off values of 0.3 μg/mL (Snibe) and 5.0 AU/mL (Medcaptain), respectively. Consequently, most subjects became seropositive within the mRNA vaccine group (88% and 98%), while less individuals showed seropositivity following adenoviral vector (68% and 86%), and inactivated virus vaccinations (56% and 64%, respectively) (Figure 2A–F). 

In parallel, total S-Ab levels were re-evaluated in each vaccination sub-group when considering the cut-off values of both neutralization tests, and we set a new cut-off value for Roche S-Abs as of 166 BAU/mL in contrast to the manufacturer’s original cut-off value of 0.8 BAU/mL. Accordingly, a similar proportion of seropositivity was observed in mRNA (86%), adenoviral vector (58%) and inactivated whole-virus vaccines (40%) based on either N-Ab cut-off value (Figure 3A–F). 

Moreover, the diagnostic usefulness of this new cut-off value of S-Abs for discrimination of seropositivity following two doses of vaccines was examined by ROC-AUC curve analysis. This showed a substantial AUC value of 0.974 (*p* < 0.0001) of S-Ab levels with 95% sensitivity and 85% specificity according to Snibe results, while a very similar AUC was determined (0.976; *p* < 0.0001) with 96% sensitivity and 73% specificity using Medcaptain data (Figure 4A,B). These results indicate the difference in immunogenicity induced by various vaccinations but also underline the reliable clinical application of these SARS-CoV-2 serology tests.

### 3.4. Analysis of Concordance among Serological Tests to Detect Seropositivity after Three Types of COVID-19 Vaccines

To explore the agreement among the three different serological assays to indicate seropositivity after immunization, we determined Cohen’s kappa coefficients for all tests in each vaccination subgroup and within the whole population (Table 2). Considerable agreements were found for all methods, especially between the two neutralization tests in the entire cohort (86.7%) achieving a high kappa coefficient of 0.635. When the concordance was further analyzed in each vaccination sub-group, the highest agreement percentage value was shown among subjects with mRNA vaccines between Snibe-Roche (94%), Medcaptain-Roche (88%) and Snibe-Medcaptain (90%) measurements (Table 2). These findings correlate with our aforementioned data above. 

### 3.5. Follow-Up of Vaccinated Subjects for the Risk of SARS-CoV-2 Infection

Finally, we were curious whether any of these subjects became infected with SARS-CoV-2 despite the two doses of vaccines and differently elevated neutralizing antibody titers in the next 6 months of the second dose. Based on their longitudinal anamnesis, only eight participants became SARS-CoV-2 positive proved by SARS-CoV-2 RT-PCR test with moderate COVID-19 symptoms in the entire group whose post-vaccination N-Ab levels remained low with a median value of 0.25 μg/mL or 7.28 AU/mL approaching the official cut-off values (Figure 4C,D). Furthermore, these patients had relatively small total S-Ab concentrations as well with a median value of 134 BAU/mL (Figure 4E). Taken together, there was a strong relationship between the prevention of COVID-19 illness and the presence of successful immunogenicity induced by two doses of vaccines which was effectively evaluated via these serological measurements.

## 4. Discussion

In this study, evoked humoral responses by five COVID-19 vaccines including the third-generation BNT162b1 and mRNA-1273, the second-generation Ch-AdOx1 and Gam-COVID-Vac and the first-generation BBIBP-CorV were evaluated in a SARS-CoV-2 naïve population. Vaccine coverage has been quite diverse worldwide, in fact the various types of vaccines have been applied at different ratios. There have been only a few clinical studies on humoral responses, especially outside Europe, where immunogenicity after all main types of COVID-19 vaccine could be investigated [13,14,15,16,17,18,19,20]. The Hungarian vaccination campaign was conducted with five different vaccines during the third wave of the COVID-19 pandemic in 2021 [24]. In this single-center study, we here compared immunogenicity after mRNA, adenoviral-vector-based and inactivated whole-virus vaccines via detecting humoral responses in a European population using two (Snibe and Medcaptain) SARS-CoV-2 neutralization tests parallelly.

Serum samples from age- and gender-matched healthy volunteers vaccinated with the aforementioned vaccines were collected. We observed that both mRNA-based vaccines induced higher levels of N-Ab and total S-Ab levels in contrast to adenovirus vector and inactivated whole-virus immunizations. Similar results were previously obtained showing that BNT162b2 and mRNA-1273 induced a significantly stronger response of spike-specific Ig titers than ChAdOx16 [25], or Gam-COVID-Vac [26], and BBIBP-CorV [26,27] or inactivated whole-virus vaccine Coronavac-Sinovac [28]. Importantly, the production of neutralizing antibodies against SARS-CoV-2 was also significantly stronger after vaccination with mRNA vaccines when compared to the other vaccines [13,15,17,22]. Similarly, the ACE2 blocking antibody levels were the highest to lowest in those with Moderna > Sputnik V/AZD1222 (had equal levels) > Sputnik light > Sinopharm [19]. In other European cohorts, mRNA vaccines were the most immunogenic after two doses, while the lowest antibody and neutralization potential were observed in the Sinopharm or Johnson and Johnson vaccines [20]. Furthermore, we proved that the values of these N-Abs measured by two different assays highly correlated with each other in the entire study population, which was in an agreement with the data of a recent publication [21]. Next, we analyzed the effect of age and gender on the titers of N-Ab and S-Ab levels, and no strong relationship was observed with either form of anti-spike antibodies. Other groups recently reported that total anti-SARS-CoV-2 S Ig titers did not correlate with age [29], while N-Abs showed contradictory findings, as those levels determined with plaque-reduction neutralization test (PRNT) did not correlate with age and gender; however, the results on Snibe and Medcaptain neutralization tests showed some difference between genders [23,30]. Nevertheless, since there was no difference in age and gender among our study sub-cohorts, we could compare the effects of all studied COVID-19 vaccines excluding such potential confounders. 

We then determined the ratio of seropositive subjects in each cohort to further characterize the immunogenicity of vaccines. Seropositivity was evaluated based on the manufacturer’s N-Ab cut-off value of 0.3 μg/mL and 5.0 AU/mL, respectively. Most subjects (88/98%) became seropositive within the mRNA vaccine group, while less individuals showed seropositivity following adenoviral vector, and inactivated whole-virus vaccinations (68/86% and 56/64%, respectively) depending on which neutralization test was used. These data correlated with the results of a large longitudinal study, where vaccine immunogenicity of adenoviral vector and inactivated whole-virus vaccines was investigated after two doses calculated by adjusted odds ratio as follows: rAd26-rAd5 (0.36), ChAdOx1 nCoV.19 (0.32) and BBIBP-CorV (0.56) [13]. Similarly, 75% of individuals after two doses of vaccines showed N-Ab positivity based on the Medcaptain test [23]. Our results were also supported by Cohen’s kappa coefficients where considerable agreements were found for all methods, especially between the two neutralization tests in the entire cohort (86.7%) achieving a high kappa coefficient of 0.635.

In addition, the titers of Roche S-Abs were re-evaluated based on the seropositivity results assessed by the Snibe and Medcaptain assays, and we set a new cut-off value of total spike Ig as of 166 BAU/mL. The diagnostic usefulness of S-Abs with this new cut-off value was examined for the discrimination of seropositivity by ROC curve analysis that showed a substantial AUC value of 0.975 considering the results of both neutralization tests. Optimal thresholds were also tested by Padoan et al. who found similar results on Roche total spike antibody (cut-off value of 205 BAU/mL with an AUC value of 0.990) and Snibe neutralizing antibody (cut-off of 0.27 μg/mL with an AUC of 0.981) [30]. When Liu et al. investigated the cut-off value of N-Abs measured by the Medcaptain assay, 6.43 AU/mL was determined with 78% sensitivity and 87% specificity [23]. For neutralizing antibody, we used the manufacturer’s threshold (0.3 μg/mL or 125 BAU/mL based on the official conversion factor to transform μg/mL into BAU/mL) to detect seropositivity that was close to our new threshold of anti-SARS-CoV-2 S total Ig levels (166 BAU/mL).

On the other hand, we analyzed its utility of functional antibodies to predict the risk of SARS-CoV-2 infection. It has been described that the half-life of N-Abs after vaccines was 10–11 weeks [23], while others reported high neutralizing bioactivity maintained at least 6 months after vaccination, and the mean values of anti-RBD IgG showed a marked decline after 6 months [31]. Hence, we followed up these subjects for the next 6 months of their second dose whether they became infected by SARS-CoV-2. Only eight participants became SARS-CoV-2 positive with moderate COVID-19 symptoms in the entire group whose post-vaccination N-Ab levels remained low with a median value of 0.25 μg/mL or 7.28 AU/mL that underlines the reliability of the thresholds of the Snibe and Medcaptain neutralization tests. The direct relationship between neutralizing antibody titers and protection against COVID-19 has been confirmed in recent meta-analyses [32,33]. 

The main limitation of the study is the relatively small number of subjects, because we studied a selected population during the pandemic and the vaccination campaign who needed to be matched with each other in age, gender, elapsed time, and negative history for previous SARS-CoV-2 infection among different vaccination groups. Due to the high incidence of SARS-CoV-2 infection during the study period, it was difficult to collect many COVID-19 naïve participants who gained the same type of mRNA or adenoviral vector-based vaccines. Consequently, we merged subjects with BNT162b2 and mRNA-1273, as well as those with ChAdOx1 and Gam-COVID-Vac into one vaccine group. Importantly, these two-dose immunizations in case of all subjects were homologous, such as ChAdOx1/ChAdOx1. However, the strength of our work is that this is a “real-life” clinical study to reflect aspects of routine care using two different automated neutralization tests. A second limitation of this study was that we did not use any surrogate virus neutralization tests (sVNT), such as Genscript (cPass) to further characterize the Snibe and Medcaptain neutralization assays in this study cohort. Notably, both N-Ab assays were previously investigated by the manufacturers (Snibe and Shenzhen Medcaptain Medical Technology Co.) validating the efficacy of these tests. Moreover, when four sVNT assays were compared with the conventional VNT and neutralizing antibody levels determined by Snibe test showed a 98.8% overall agreement with the PRNT [34], while the results of Medcaptain test were highly correlated with microneutralization test [23].

## 5. Conclusions

Both Snibe and Medcaptain SARS-CoV-2 N-Ab automated assays are effective to evaluate humoral responses after the administration of various COVID-19 vaccines. Compared to the adenovirus vector and inactivated whole-virus vaccines, the mRNA COVID-19 vaccines generate more robust SARS-CoV-2 N-Ab titers assessed by two distinct tests, which highly correlate with each other and Roche total S-RBD Ig levels.

## Figures and Tables

**Figure 1 microorganisms-11-01187-f001:**
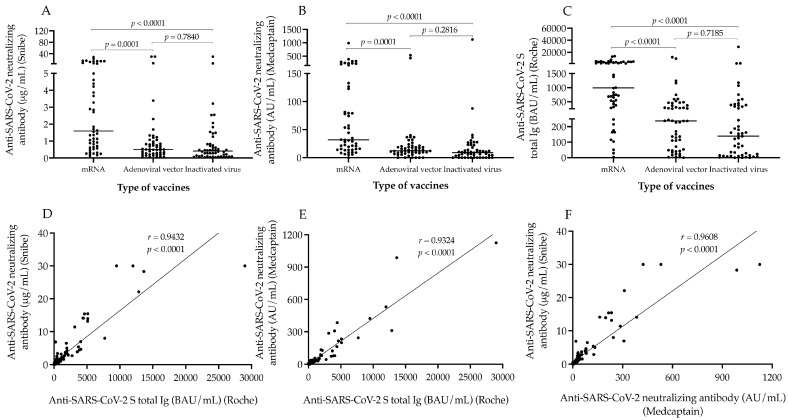
Analysis of the levels of SARS-CoV-2 neutralizing antibody and anti-SARS-CoV-2 S total Ig after different types of vaccines. The figure shows the levels of SARS-CoV-2 neutralizing (**A**,**B**) and anti-SARS-CoV-2 S total antibody (**C**) among individuals who received two doses of mRNA, adenoviral-vector-based and inactivated whole-virus vaccines. The correlation between the values of neutralizing and total S-Ab was also studied by Spearman test involving both all data (**D**–**F**). Median values of antibody titers are depicted with solid line. Wilcoxon matched pairs signed rank test was used to compare data between two groups. mRNA vaccines: BNT162b2, Pfizer/BioNTech and mRNA-1273, Moderna; Adenoviral vector vaccines: ChAdOx1, AstraZeneca and Gam-COVID-Vac, Sputnik V; Inactivated virus vaccine: BBIBP-CorV, Sinopharm; BAU/mL: binding antibody units per milliliter; AU/mL: arbitrary units per milliliter.

**Figure 2 microorganisms-11-01187-f002:**
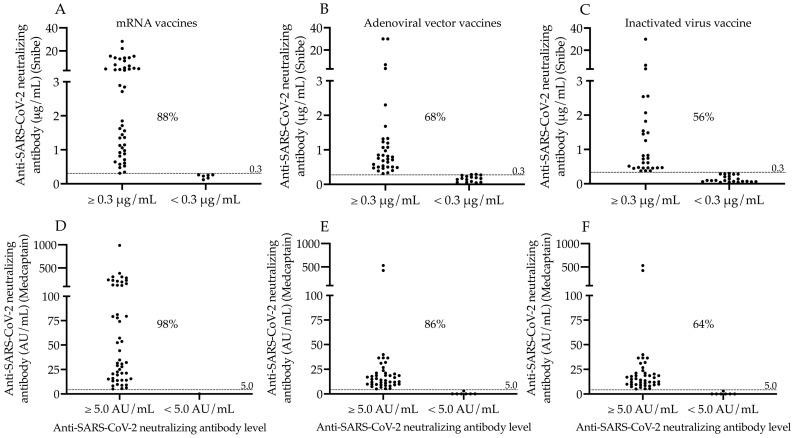
Ratio of seropositive subjects based on neutralizing antibody levels determined by the two N-Ab assays. The ratio of seropositive subjects was analyzed based on neutralizing antibody levels using two different tests after two doses of mRNA (**A**,**D**), adenoviral-vector-based (**B**,**E**) and inactivated virus vaccines (**C**,**F**). We used the cut-off value for neutralizing antibodies (0.3 μg/mL of Snibe (**A**–**C**) and 5.0 AU/mL of Medcaptain assay (**D**–**F**) to sub-group the study participants. The percentage values of seropositive individuals determined by either serology test are depicted in each sub-cohort. mRNA vaccines: BNT162b2, Pfizer/BioNTech and mRNA-1273, Moderna; Adenoviral vector vaccines: ChAdOx1, AstraZeneca and Gam-COVID-Vac, Sputnik V; Inactivated virus vaccine: BBIBP-CorV, Sinopharm; AU/mL: arbitrary units per milliliter.

**Figure 3 microorganisms-11-01187-f003:**
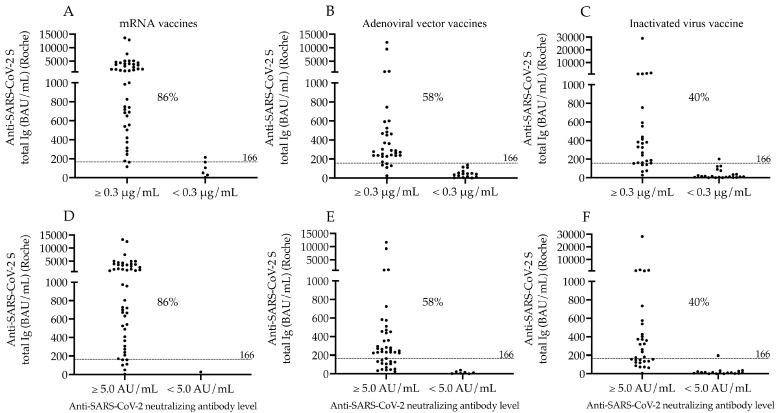
Ratio of seropositive subjects in different vaccination groups according to anti-SARS-CoV-2 S total Ig levels when N-Abs results are also considered in the same samples. The percentage of seropositive subjects in each vaccination group was reevaluated based on the Roche anti-SARS-CoV-2 S total Ig levels after two doses of mRNA (**A**,**D**), adenoviral-vector-based (**B**,**E**) and inactivated virus vaccines (**C**,**F**) using its new cut-off value (166 BAU/mL) when the results of Snibe (**A**–**C**) and Medcaptain (**D**–**F**) neutralization tests were also considered to classify subjects. The percentage values of seropositive individuals are depicted in each sub-cohort. mRNA vaccines: BNT162b2, Pfizer/BioNTech and mRNA-1273, Moderna; Adenoviral vector vaccines: ChAdOx1, AstraZeneca and Gam-COVID-Vac, Sputnik V; Inactivated virus vaccine: BBIBP-CorV, Sinopharm; AU/mL: arbitrary units per milliliter.

**Figure 4 microorganisms-11-01187-f004:**
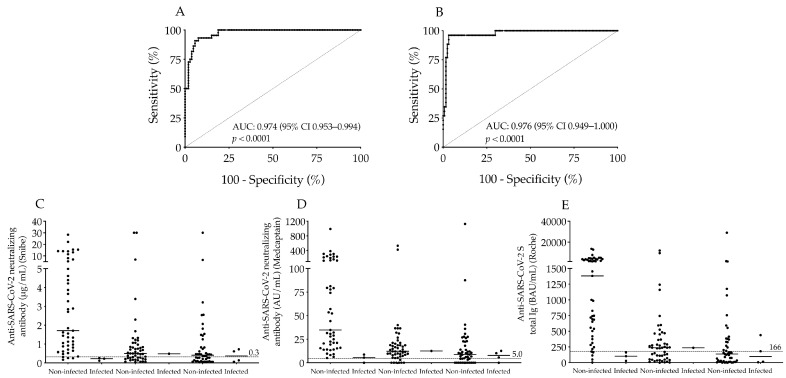
Analysis of the new cut-off value of the Roche anti-SARS-CoV-2 S total Ig test and the ratio of SARS-CoV-2 infected individuals after vaccination within 6 months. The ROC-AUC curve analysis of the new cut-off value (166 BAU/mL) of Roche anti-SARS-CoV-2 S total Ig to discriminate seropositivity in association with Snibe N-Abs (**A**) and Medcaptain N-Abs (**B**). We also calculated how effective these COVID-19 vaccines were to prevent COVID-19 illness in the next 6 months after the second dose in each sub-group via the post-vaccination neutralizing (**C**,**D**) and anti-SARS-CoV-2 S total Ig levels (**E**). Using set cut-off values of both antibodies indicated with dashed line we found low titers of both types of N-Abs and S-Abs in those who were infected by SARS-CoV-2 within the study period (**C**–**E**). Median values of antibody titers are depicted with solid line. AUC: area under curve; AU/mL: arbitrary units per milliliter; BAU/mL: binding antibody units per milliliter.

**Table 1 microorganisms-11-01187-t001:** Main demographical characteristics of recruited study participants in the three types of COVID-19 vaccination.

Characteristics	mRNA Vaccines (*n* = 50)	Adenoviral Vector Vaccines (*n* = 50)	Inactivated Whole-Virus Vaccine (*n* = 50)
Age (years), median (IQR)	61.1 (48.9–70.3)	53.6 (47.8–62.4)	63.6 (46.7–66.5)
Sex (female/male), *n*/*n*	35/15	26/24	30/20
Elapsed time between 2nd dose and serology (days), median (IQR)	36 (22–62)	36 (25–53)	47 (21–67)
Hypertension, *n*	24	24	29
Cardiovascular disease, *n*	6	5	4
Diabetes mellitus, *n*	6	6	9
Chronic renal disorder, *n*	5	7	6
Autoimmune disease, *n*	4	4	3
Lung disease, *n*	1	1	2
Thyroid disorder, *n*	2	3	2

Mann–Whitney U-test was used to compare mRNA vs. adenoviral vector, mRNA vs. inactivated whole-virus, and adenoviral vector vs. inactivated whole-virus vaccination sub-groups for age (*p* = 0.0694, *p* = 0.2487 and *p* = 0.0923, respectively), gender (*p* = 0.0657, *p* = 0.2945 and *p* = 0.4203, respectively) and elapsed time interval (*p* = 0.5745, *p* = 0.3670 and *p* = 0.0726, respectively). mRNA vaccines: BNT162b2, Pfizer/BioNTech and mRNA-1273, Moderna; Adenoviral vector vaccines: ChAdOx1, AstraZeneca and Gam-COVID-Vac, Sputnik V; Inactivated whole-virus vaccine: BBIBP-CorV, Sinopharm; *n* = number of patients/samples.

**Table 2 microorganisms-11-01187-t002:** Analysis of concordance among serological tests to detect seropositivity after three types of COVID-19 vaccination using Cohen’s kappa coefficients.

Comparisons	Agreement	Cohen’s Kappa [95% CI]	*p* Value
Snibe-Roche (all vaccination groups)	87.3%	0.722 [0.566–0.878]	<0.0001
Snibe-Roche (mRNA)	94.0%	0.735 [0.458–1.000]	<0.0001
Snibe-Roche (adenoviral vector)	88.0%	0.749 [0.481–1.000]	<0.0001
Snibe-Roche (inactivated whole-virus)	80.0%	0.609 [0.346–0.872]	<0.0001
Medcaptain-Roche (all vaccination groups)	76.7%	0.458 [0.320–0.595]	<0.0001
Medcaptain-Roche (mRNA)	88.0%	0.229 [0.048–0.397]	0.0061
Medcaptain-Roche (adenoviral vector)	70.0%	0.343 [0.134–0.552]	0.0006
Medcaptain-Roche (inactivated whole-virus)	72.0%	0.469 [0.222–0.716]	0.0001
Snibe-Medcaptain (all vaccination groups)	86.7%	0.635 [0.483–0.785]	<0.0001
Snibe-Medcaptain (mRNA)	90.0%	0.261 [0.073–0.447]	0.0031
Snibe-Medcaptain (adenoviral vector)	82.0%	0.514 [0.271–0.756]	<0.0001
Snibe-Medcaptain (inactivated whole-virus)	88.0%	0.752 [0.478–1.000]	<0.0001

mRNA vaccines: BNT162b2, Pfizer/BioNTech and mRNA-1273, Moderna; Adenoviral vector vaccines: ChAdOx1, AstraZeneca and Gam-COVID-Vac, Sputnik V; Inactivated whole-virus vaccine: BBIBP-CorV, Sinopharm.

## Data Availability

All data are contained within the article.

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
