# Peer review of "Evaluation of the Diagnostic Performance of Two Automated SARS-CoV-2 Neutralization Immunoassays following Two Doses of mRNA, Adenoviral Vector, and Inactivated Whole-Virus Vaccinations in COVID-19 Naïve Subjects"

_microorganisms, 2023, doi:10.3390/microorganisms11051187_

Round 1

Reviewer 1 Report

The manuscript titled "Evaluation of the diagnostic performance of two automated SARS-CoV-2 neutralization immunoassays following two doses of mRNA, adenoviral vector, and inactivated whole-virus vaccinations in COVID-19 naïve subjects" presents an important contribution to the understanding of humoral responses after different types of COVID-19 vaccines. The study evaluates the anti-SARS-CoV-2 neutralizing antibody titers using two different neutralization assays and compares them to total spike antibody levels among healthy participants who received different COVID-19 vaccines.

How were the participants chosen? Randomly? Why more female participants than males are in all three main types of COVID-19 vaccination?

Out of 150 participants, 50 received mRNA vaccines (n = 39 with BNT162b2, Pfizer/BioN- 89 Tech and n = 11 with mRNA-1273, Moderna), 50 individuals were injected with adenoviral 90 vector vaccines (n = 28, ChAdOx1, AstraZeneca and n = 22, Gam-COVID-Vac, Sputnik V). Did the authors consider the differences between vaccines from different company? Does this difference have any impact on the measurement results?

The results of this study are interesting and provide important insights into the effectiveness of different COVID-19 vaccines. However, there are several areas that require clarification and further explanation. Firstly, there is a need for more detail on the methods used to select the healthy participants enrolled in the study. Additionally, it would be beneficial to provide a more comprehensive discussion on how the findings of this study compare to previous research on the topic. Furthermore, while the authors acknowledge the limitations of the study, it would be helpful to discuss the implications of these limitations on the generalizability of the results. For instance, the relatively small sample size and the absence of surrogate virus neutralization tests (sVNT) could impact the reliability of the study's findings. Therefore, it is recommended that the authors include a more thorough discussion of these limitations. Overall, the manuscript provides a valuable contribution to the field of COVID-19 vaccination, but some improvements are necessary before publication. Therefore, I would recommend minor revisions to address the issues mentioned above.

Reviewer 2 Report

Overall

The manuscript entitled: "Evaluation of the diagnostic performance of two automated SARS-CoV-2 neutralization immunoassays following two doses of mRNA, adenoviral vector, and inactivated whole-virus vaccinations in COVID-19 naïve subjects” submitted by Csoma et al reports the immunogenicity comparison of three different COVID-19 vaccines in a Hungarian cohort through automated SARS-CoV-2 neutralization immunoassays, besides they evaluated the two immunoassays which results showed very similar performance.

Major comments

-Lines 71-74 The authors stated: “Consequently, less is known about the effect of all available types of vaccine and, to the best of our knowledge, there have been only a few studies on immune activation, where immunogenicity after two doses of all main types of vaccine was investigated, such as in Argentina and Bangladesh [10-13]”. However, this reviewer performed a quick search excluding heterologous vaccination protocols, I found the following two articles which had the same objective as the present article, and it is very likely that there are more, therefore, I suggest that the authors explore the state of the art deeply, reword their claim, and add the other relevant references.

Jeewandara C, et al.  Comparison of the immunogenicity of five COVID-19 vaccines in Sri Lanka. Immunology. 2022 Oct;167(2):263-274. doi: 10.1111/imm.13535.

Adjobimey T, et al. Comparison of IgA, IgG, and Neutralizing Antibody Responses Following Immunization With Moderna, BioNTech, AstraZeneca, Sputnik-V, Johnson and Johnson, and Sinopharm's COVID-19 Vaccines. Front Immunol. 2022 Jun 21;13:917905. doi: 10.3389/fimmu.2022.917905.                                                                    

-Line 383-385. The authors claimed that in this study, the efficacy of the mRNA, adenoviral vector-based, and inactivated whole-virus COVID vaccines were compared. If they would have added an explanation about the correlation between neutralizing antibody titers with protection against COVID-19, this would have supported their asseveration (e.g. the studies published by Khoury et al and Cromer D, et al.), however, the authors used in the manuscript ”efficacy” as a synonym for immunogenicity (lines 242, 317, and 384) which is a wrong assumption. The efficacy concept is the level of protection in a population afforded by a vaccine, and an example of this kind of study is reported by Baden et al.; thus this study is not an efficacy study.   

Baden LR, et al. Efficacy and Safety of the mRNA-1273 SARS-CoV-2 Vaccine. N Engl J Med. 2021 Feb 4;384(5):403-416. doi: 10.1056/NEJMoa2035389.

Correlation neutralizing antibodies and protection

Khoury DS, et al. Neutralizing antibody levels are highly predictive of immune protection from symptomatic SARS-CoV-2 infection. Nat Med. 2021 Jul;27(7):1205-1211. doi: 10.1038/s41591-021-01377-8.

Cromer D, et al. Neutralising antibody titres as predictors of protection against SARS-CoV-2 variants and the impact of boosting: a meta-analysis. Lancet Microbe. 2022 Jan;3(1):e52-e61. doi: 10.1016/S2666-5247(21)00267-6.

-Line 189. I guess the best word that fits better in the context is “performance” instead of “efficacy”.

Minor comments

The Figure legends should be at the bottom of the Figures instead of the top.

Reviewer 3 Report

General considerations about the manuscript:

The focus of the article should be the evaluation of the diagnostic performance of the two automated SARS-CoV-2 neutralization immunoassays, but confusing and not applicable elements, such as the evaluation of vaccine effectiveness, are added.

In general, the article is not clearly written. 

The study design is totally missing and all the topics appear confused and disorganized.

The content of the manuscript does not comply with the correct structure of a scientific article, indeed some contents are found in sections other than those intended.

The manuscript, overall, does not add data to the existing literature and it refers to the very first phase of vaccination against SARS-Cov-2 (primary cycle).

Moreover, the conclusive results concerning the enhanced antibody response that develops following vaccination with mRNA vaccines compared to the adenovirus vector and inactivated whole-virus vaccines are widely known and reported in literature. 

  • ABSTRACT : 

Authors attribute the importance of this work to the lack of similar literature data; this could have been true at the dawn of the COVID-19 vaccination campaign, but to date the scientific literature is teeming with data relating to the topic, often deriving from much more structured and accurate analyses.

Already in the abstract section you can find technical inaccuracies that could invalidate the collected data (example: line 36).

  • INTRODUCTION:

  • LINES  50-51: please explain the choice of data source (see bibliographic source indicated): why are the authoritative data of the WHO dashboard not used?

  • LINE 70: bibliographic source n'7 is not strictly pertinent

  • LINES 71-73:  the reported data appear obsolete

  • MATERIALS AND METHODS:

  • LINE 88: please explain how the serologic evidence of no prior SARS-CoV-2 infection has been determined (were participants tested for antibodies against N protein?)

  • LINES 89-92 : subjects vaccinated with different vaccines were categorized into the same group according to vaccine technology, but this could represent a bias source of evaluation error.
  • a demographic description of the sampled subjects should be included in this section.

  • LINE 94: specimens were obtained “41 (22—65) days after their second dose in the whole population”: the interval between the vaccination date and the collection of the samples is too variable (from 20 days to more than two months depending on the subject). The study design should have provided for a narrower time range (fixed and pre-established) regarding the execution of the blood sample.

  • LINES 95-97: this information would go into the results section.

  • LINE 130: the Elecsys® Anti-SARS-CoV-2 S assay is an immunoassay for the in vitro quantitative determination of antibodies (IgG and IgM) against the spike protein (S) receptor binding domain of SARS-CoV-2 (RBD) in human serum and plasma. The data sheet of the assay does not refer to the detection of anti S1 antibodies, please better specify the target examined. [https://diagnostics.roche.com/it/it/products/params/elecsys-anti-sars-cov-2-s.html#productInfo]

  • LINES 139-140: it is not scientifically correct to compare results expressed in different units (µg/mL, AU/mL and BAU/mL).

  • materials and methods used for the results reported in paragraphs 3.3 and 3.4 are not mentioned.

  • RESULTS:

  • LINES 170-177: it is not scientifically correct to compare methods that offer results in different units of measurement.

  • LINE 244 : subparagraph not related to the subject of the study.

  • LINE 294: please describe clearly why you set the new cut-off value. The motivation is not understandable.

  • LINE 346 : subparagraph not related to the subject of the study.

  • DISCUSSION:

The purpose of the study and the description of the vaccine's distinct mechanisms to induce immune activation are reported in the discussion section; they should be stated instead in the introduction (although reporting the mechanism to induce immune activation may not be necessary in such a study).

  • CONCLUSIONS  :

  • the conclusions offered by this study are obsolete and not scientifically meaningful due to poor methodological rigor .

Round 2

Reviewer 1 Report

The current version is ok for publication.

Author Response

We thank the Reviewer for the favorable evaluation.

Reviewer 2 Report

I only have a couple of observations, first, probably the edition software moved the Figure 1, because the legend figure is far away, second, I’m get used to see all figure legends (including short titles) at the bottom of the figures, I assume the right way will be placed by the editorial.

Author Response

We thank the Reviewer for the favorable evaluation. Based on your suggestion, we have inserted the title of the figures and tables under the figures as a part of the figure legends. The structure of the article has been double-checked.